# NMR Experiments Shed New Light on Glycan Recognition by Human and Murine Norovirus Capsid Proteins

**DOI:** 10.3390/v13030416

**Published:** 2021-03-05

**Authors:** Robert Creutznacher, Thorben Maass, Patrick Ogrissek, Georg Wallmann, Clara Feldmann, Hannelore Peters, Marit Lingemann, Stefan Taube, Thomas Peters, Alvaro Mallagaray

**Affiliations:** 1Institute of Chemistry and Metabolomics, University of Lübeck, Ratzeburger Allee 160, 23562 Lübeck, Germany; r.creutznacher@uni-luebeck.de (R.C.); t.maass@uni-luebeck.de (T.M.); patrick.ogrissek@student.uni-luebeck.de (P.O.); gewaelbe@googlemail.com (G.W.); clara.feldmann@student.uni-luebeck.de (C.F.); hanne.pepo@gmail.com (H.P.); alvaro.mallagaraydebenito@uni-luebeck.de (A.M.); 2Institute of Virology and Cell Biology, University of Lübeck, Ratzeburger Allee 160, 23562 Lübeck, Germany; marit.lingemann@gmail.com

**Keywords:** norovirus P-domain, chemical shift perturbation, histo blood group antigens, sialoglycans, protein-carbohydrate recognition

## Abstract

Glycan–protein interactions are highly specific yet transient, rendering glycans ideal recognition signals in a variety of biological processes. In human norovirus (HuNoV) infection, histo-blood group antigens (HBGAs) play an essential but poorly understood role. For murine norovirus infection (MNV), sialylated glycolipids or glycoproteins appear to be important. It has also been suggested that HuNoV capsid proteins bind to sialylated ganglioside head groups. Here, we study the binding of HBGAs and sialoglycans to HuNoV and MNV capsid proteins using NMR experiments. Surprisingly, the experiments show that none of the norovirus P-domains bind to sialoglycans. Notably, MNV P-domains do not bind to any of the glycans studied, and MNV-1 infection of cells deficient in surface sialoglycans shows no significant difference compared to cells expressing respective glycans. These findings redefine glycan recognition by noroviruses, challenging present models of infection.

## 1. Introduction

Noroviruses are single-stranded (+)−RNA viruses associated with acute gastroenteritis in mammalian hosts. In fact, human noroviruses (HuNoV) are the leading cause for viral gastroenteritis worldwide, with the majority of norovirus outbreaks since 2012 having been caused by genogroup II, genotype 4 (GII.4) viruses [1,2,3]. Histo-blood group antigens (HBGAs) have been identified as critical structural determinants for HuNoV infection [4,5,6]. The host enzyme α-1,2-fucosyltransferase (FUT2) is essential for HBGA biosynthesis. A functional FUT2 enzyme defines the “secretor” phenotype, where HBGAs are present on the surface of intestinal epithelial cells and become secreted. Individuals lacking FUT2 activity are “non-secretors” and highly resistant against many HuNoV strains. Noroviruses bind HBGA at the interface of a dimeric protruding (P)-domain of the viral capsid protein VP1. Despite an increasing number of high-resolution crystal structures that describe the interactions of HBGAs with HuNoV P-domains at atomic resolution, it has remained enigmatic as to how this interaction with HBGAs promotes infection [7,8].

There are exceptions, where HBGA binding could not be determined [9], or mutations notably modulated HBGA binding [10], or expression of FUT2 in permissive cell lines, which is essential for HBGA biosynthesis in continuous cell lines, is not sufficient to initiate infection [11]. Recently, a study in human intestinal enteroids clearly demonstrated the necessity of functional FUT2 for HuNoV infection [12]. The same study describes that a GII.3 strain was able to replicate in secretor-negative human intestinal enteroids in the presence of the bile acid glycochenodeoxycholic acid (GCDCA). Therefore, it is likely that additional factors, possibly even a yet unknown proteinaceous entry receptor, as recently identified for murine noroviruses (MNV) [13,14,15,16], contribute to HuNoV infections.

In addition to neutral HBGAs, other glycans have been shown to contribute to calicivirus infections. For instance, feline caliciviruses and Tulane virus were shown to require sialic acids for binding and infection [17,18]. MNVs share many biological features with HuNoV, such as the intestinal phenotype and fecal oral spread. However, unlike HuNoV they can be easily studied in cell culture and in their native small animal host. It has been shown that the interaction of MNV-1 with murine macrophages was sensitive to neuraminidase treatment and competition with sialic acid specific lectins [19]. Strain specific differences in the ability to bind sialic acid and mutational analysis further suggested a sialic acid binding site [20] in analogy to the HBGA binding sites of GII.4 P-domains of HuNoV. However, structural data for MNV P-domain complexed with glycans are still absent. *Bona fide* receptors, murine CD300lf or CD300ld, have recently been shown to be essential and sufficient for MNV entry [15,16], and high-resolution crystal structures for the MNV P-domain complexed with CD300lf are available [13,14].

Binding of sialoglycans to HuNoV GII.4 P-domains has been reported using different biophysical techniques [21,22]. Using native mass spectrometry, dissociation constants have been determined for binding of ganglioside-derived glycans to HuNoV GII.4 VA387 P-dimers [22]. Likewise, binding of 3′-sialyllactose to HuNoV GII.4 MI001 P-dimers has been described based on native mass spectrometry, STD NMR, and SPR experiments [21].

A number of studies have addressed the specific interaction between HuNoV GII.4 P-domains (P-dimers) and HBGAs. High-resolution crystal structures have provided a clear picture of P-dimer binding pockets for HBGAs (for a recent review see [23,24]. Consistent with these results, STD NMR experiments have demonstrated that L-fucose is the minimal recognition element required for specific binding [25]. Binding affinities were obtained from STD NMR titrations [21,26,27,28], and affinities and stoichiometries were provided by mass spectrometry experiments [21,27,29,30,31], reporting dissociation constants *K_D_* in the μM to low mM range. For the interaction of virus particles with host cells multivalency has to be considered, eventually leading to firm attachment of viruses to plasma membranes. This subject has been studied in detail by using total internal reflection (TIRF) microscopy [32] and quartz microbalance dissipation monitoring [33]. Notably, it has been recognized that different spatial presentation of HBGAs as in type-1 vs. type-2 glycosphingolipids embedded in phospholipid membranes significantly modulates binding avidity [34]. Clearly, presentation of glycans in membranes adds an additional layer of complexity and is relevant for biological function.

Apart from that, it is still important to have access to reliable quantitative glycan binding data. Recent research [35] suggests that existing reports are inconsistent, calling for a systematic reevaluation. Therefore, we set out to collect a comprehensive dataset for binding of glycans to human and murine norovirus capsid proteins based on protein- as well as ligand-based NMR studies.

## 2. Materials and Methods

### 2.1. Protein Biosynthesis and Purification

Recombinant P-domain proteins from GII.4 NoV strains Saga 2006 (GenBank ID: AB447457, aa 225–530), MI001 (KC631814, aa 225–530), and VA387 (AY038600, aa 225–529) were synthesized recombinantly and purified as described previously [35,36]. Due to the purification strategy, the amino acids GPGS were added N-terminally of the native VP1 sequence to yield an enzymatic cleavage site. [*U*-^2^H,^15^N]-labeled protein samples were denatured and refolded to complete the HD back-exchange after expression in deuterated minimal media. Protein species with different deamidation status of N373 were separated using cation exchange chromatography (IEX) with a 6 mL Resource S column (GE Healthcare, Chicago, IL, USA). Samples were applied onto the column in 20 mM sodium acetate buffer (pH 4.9) and eluted with a linear salt gradient up to 500 mM NaCl. Deamidation kinetics of P-domains from strains Saga 2006 and MI001 were obtained by incubation of protein aliquots (1.5 and 0.6 mg mL^−1^, respectively) in 75 mM sodium phosphate buffer, 100 mM NaCl (pH 7.3) at 298 K. Samples were subjected to IEX separation at selected time point and peaks in the UV chromatogram at 214 nm were integrated using Unicorn 7 software (GE Healthcare, Chicago, IL, USA). The decay in N/N dimers was fitted to an exponential decay model and used to derive N373 half-life times. Stability of the N373D Saga 2006 P-domain was assessed by incubation of purified protein aliquots with a concentration of 1.3 mg mL^−1^ in 75 mM sodium phosphate buffer, 100 mM NaCl (pH 7.3) for 382 h at 298 K and subsequent IEX analysis. Synthesis and purification of MNV P-domains from strains MNV07 (AET79296, aa 228–530, N-terminal addition of GPGS peptide) and CW1 (DQ285629, aa 228–530, N-terminal addition of GP peptide) followed the same strategy as for GII.4 P-domains with the only difference being the running buffer used in the final size exclusion chromatography step (20 mM sodium acetate, 100 mM NaCl, pH 5.3). GII.10 Vietnam 026 VLPs (GenBank ID: AF504671) [37] were a gift from Dr. Grant Hansman (University of Heidelberg and the DKFZ, Heidelberg, Germany).

### 2.2. Enzymatic Synthesis of Blood Group B-Trisaccharide (3)

Human Galactosyltransferase B (GTB) was overexpressed from *E. coli* as previously reported [38] using an optimized purification protocol [39]. Briefly, GTB was eluted from the UDP-affinity column with a 10 mM EDTA solution. Fractions showing UV activity (280 nm) were pooled and dialyzed against 2 L of 50 mM MOPS pH 6.8, 1 mM DTT, 5 mM MgCl_2_ and 100 mM NaCl at room temperature. GTB was stored at 1.2 mg/mL protein concentration at 4 °C for no longer than two months. GTB enzymatic activity was measured using radiolabeled uridine-diphospho-α-D-[U-^14^C]-Galactose lithium salt in 70% ethanol prior to the synthetic reaction [40], showing specific activity of 10 U/mg. For the enzymatic synthesis of **3**, the following reactants were mixed in this precise order: H-disaccharide **1** (10.5 mg, 30 µmol, 1.00 equiv.) was dissolved in 15 mL of 50 mM Bis-Tris buffer pH 6.85 at 37 °C, into which UDP-Galactose (25.48 mg, 45 µmol, 1.50 equiv.), BSA (0.15 mL of a stock solution containing 100 mg/mL BSA in 100 mM Bis-Tris buffer pH 6.85 at 37 °C, final conc.: 1 mg/mL), 17 U/mL CIAP (10,000 U/mL), GTB (0.255 mL of GTB stock solution containing 10 U/mL and 1.2 mg/mL; 170 mU/mL final GTB conc.) and MnCl_2_ (0.31 mL of a stock solution containing 1 M MnCl_2_, 20 mM final conc.,) were added. The pH of the reaction at the starting point was measured with an electrode (6.85 at 37°), and the reaction mixture was incubated at 37 °C under shaking (150 rpm) for 3–7 days. To avoid enzymatic inhibition, the pH was daily readjusted to pH > 6.7 (37 °C) by the addition of a solution containing 500 mM Bis-Tris pH 6.85 (37 °C). After the first 48 h, the reaction was supplemented with extra UDP-Galactose (8.5 mg, 15 µmol, 0.50 equiv.) and GTB (100 mU/mL). Formation of **3** was monitored by TLC (DCM:MeOH 6.5:3.5, Rf = 0.12 in DCM:MeOH 6.5:3.5). When no more H-disaccharide **1** was observed and the pH remained constant for 24 h, the reaction was quenched by the addition of 20 mM EDTA. Crude was lyophilized, dissolved in 4 mL MP-H_2_O and purified by size exclusion chromatography (Bio-Gel^®^ P2, BioRad, Hercules, CA, USA) in MP-H_2_O (column diameter 2 cm, length 100 cm, retention time ~120 min, flow 0.5 mL/min). Fractions containing **1** were pooled together, lyophilized, dissolved in 2 mL MP-H_2_O and purified by RP-HPLC (20 min MP-H_2_O with 0.01% TFA, then gradient to MP-H_2_O:MeOH 4:1 with 0.01% TFA in 60 min, retention time ~21 min, flow 9 mL/min). In both purification steps, the carbohydrate was detected by TLC. After lyophilization, B-trisaccharide **3** (10.01 mg, 19.5 µmol, 65%) was obtained as a white solid.

M.p. = 204–206 C°.—[α]**_D_^21^** = +97.69 (13.09 mg/mL, H_2_O).—IR (KBr) 3328, 2918, 2115, 1673, 1337 cm^−1^.—^1^H-NMR (500 MHz, D_2_O) δ 5.69 (d, *J* = 4.2 Hz, 1H, H_Gal1_-1), 5.20 (d, *J* = 3.5 Hz, 1H, H_Gal2_-1), 5.13 (d, *J* = 3.7 Hz, 1H, H_Fuc_-1), 4.33–4.30 (m, 2H, H_Gal2_-5, H_Gal1_-4), 4.18–4.13 (m, 2H, H_Fuc_-5, H_Gal1_-2), 4.10–4.06 (m, 2H, H_Gal1_-5, H_Gal1_-3), 3.96–3.73 (m, 10H, H_Gal2_-6_a&b_, H_Gal1_-6_a&b_, H_Fuc_-2, H_Fuc_-4, H_Gal2_-2, H_Fuc_-3, H_Gal2_-3, H_Gal2_-4), 1.26 (d, *J* = 6.5 Hz, 3H, CH_3Fuc_).—^13^C NMR (125 MHz, D_2_O) δ 103.4, 96.1, 91.2, 75.5, 75.2, 74.6, 74.4, 73.8, 72.2, 72.2, 72.2, 71.0, 70.6, 70.5, 67.2, 64.2, 64.1, 18.3.—Anal. calc. for: C_18_H_31_N_3_O_14_: C, 42.11; H, 6.09; N, 8.18. Found: C, 43.01; H, 6.08; N, 8.29.

### 2.3. NMR Spectroscopy

NMR samples were prepared in 3 mm NMR tubes with a volume of 160 µL. NMR experiments with [*U*-^2^H,^15^N]-labeled proteins were acquired with samples containing 10% D_2_O. Samples with human NoV P-domains contained 200 µM 2,2-dimethyl-2-silapentane-5-sulfonate-d_6_ (DSS-d_6_, Sigma-Aldrich) and 300 µM imidazole (Roth) to monitor the sample pH [41]. MNV P-domain samples contained 500 µM DSS-d_6_. GII.10 Vietnam026 VLP samples were prepared in PBS buffer (pH* 7.20) containing 8% D_2_O and 100 µM DSS-d_6_. Protein concentrations and sample buffers are compiled in the figure and table legends of the respective datasets or in Appendix A. Carbohydrates were obtained from sources given in Appendix A. Assignment of resonances from α-azido B-trisaccharide 3 and experimental conditions of respective NMR experiments for assignment are given in Appendix A, respectively.

Unless stated otherwise, NMR spectra were acquired at 298 K on a Bruker Avance III 500 MHz spectrometer with TCI cryo probe. For acquisition and processing of NMR spectra TopSpin v3.6 (Bruker, Billerica, MA, USA) was used. Protein signal positions and intensities were extracted with CcpNmr Analysis v2.4.2 [42]. ^1^H,^15^ N TROSY HSQC spectra were acquired with 128 ms acquisition time and a spectral window of 16 ppm in the direct dimension. In the indirect dimension, the spectral window was set to 35 ppm with 430 increments for human NoV P-domains and 256 increments for MNV P-domains. The relaxation delay was 1.5 s. The number of scans was chosen according to the protein concentration of the sample, ranging from 8 to 48 corresponding to measurement times of 1.6 to 6 h. Potential ligand molecules were titrated from highly concentrated, pH-adjusted stock solutions in the same buffer as the NMR sample to minimize dilution effects and pH-artifacts. Titrations of GII.4 NoV P-domains with HBGAs were used to obtain Euclidean Chemical Shift Perturbations (CSPs) Δν according to Equation (1).
(1)$$\Delta \nu_{eucl}=\sqrt{\Delta \nu_{H}^{2}+\Delta \nu_{N}^{2}}$$
where $\Delta \nu_{H}$ and $\Delta \nu_{N}$ are CSPs in the respective dimensions in units of Hz. The observed $\Delta \nu_{obs}$ at a given total ligand concentration *L_t_* are linked to the dissociation constant *K_D_* via Equation (2) [43].
(2)$$\Delta \nu_{obs}=\Delta \nu_{max}\frac{\left(P_{t}+L_{t}+K_{D}\right)-\sqrt{{\left(P_{t}+L_{t}+K_{D}\right)}^{2}-4P_{t}L_{t}}}{2P_{t}}$$
where $\Delta \nu_{max}$ is the maximum CSP at ligand saturation for each signal, and *P_t_* is the total monomeric protein concentration. Titration curves for global non-linear least squares fitting were selected according to the magnitude of the CSP at the highest ligand concentration: CSPs larger than the sum of the mean of all CSPs and two standard deviations were used to derive a *K_D_* value. Global fitting of significant CSP data was performed using in-house Python scripts, fit parameter errors were estimated using a bootstrapping approach with 1000 iterations. The titration of CaCl_2_ to MNV CW1 P-domains with an unknown amount of Ca^2+^ originating from the GCDCA stock solution (Sigma-Aldrich, St. Louis, MO, USA) was analyzed as follows. The initial Ca^2+^ concentration was estimated based on the minimum of squared residuals when varying c(Ca^2+^) between 0 and 1 mM before fitting binding isotherms. Finally, for global fitting of Equation (2) to the CSP data this initial Ca^2+^ concentration was added to the amount of CaCl_2_ titrated.

STD NMR spectra [44] of ligands binding to P-domains were acquired with relaxation delays of 5 s. Detailed sample conditions are compiled in Appendix A. The number of scans and the resulting experiment time were as follows: 1712 (7 h) for the experiment with the MNV07 strain and 3′SL at 277K, and 3584–4000 (14–16 h) for the remaining STD experiments as described in 3.2 of this manuscript. On-resonance frequencies and saturation pulse strengths were optimized for each individual sample in negative-control experiments without protein and were between −2 and −4 ppm. Off-resonance frequencies were set at 200 ppm. Saturation times were 2 s using a cascade of 50 ms Gaussian pulses with flip angles between 639° and 674°. STD NMR spectra of compound 13 were measured on a 600 MHz Bruker Avance III HD NMR spectrometer with an Ascend magnet and a TCI cryo probe using a flip angle of 905°. STD NMR experiments with GII.10 Vietnam 026 VLPs were acquired on a 600 MHz Bruker Avance III HD NMR spectrometer with an Ascend magnet and a TXI room temperature probe with the temperature set at 277 K and a relaxation delay of 20 s. On- and off-resonance frequencies were −4 and 200 ppm, respectively. B-trisaccharide was titrated from 1 mM to 50 mM final concentration. The number of scans ranged from 1200 to 200 (13–2 h). A 2 s pulse cascade of 50 ms Gaussian pulses was used for saturation of protein resonances and a 20 ms spin-lock filter was used for suppression of protein signals. For the Gaussian pulses the power level was set at 40 dB, corresponding to a flip angle of 675.5°.

STD amplification factors (STD AF) for selected signals were calculated according to Equation (3) with *I_off_* and *I_on_* being the signal intensities in the respective off- and on-resonance experiments:(3)$$STD\;AF=\frac{I_{off}-I_{on}}{I_{off}}\cdot \frac{L_{t}}{P_{t}}$$

### 2.4. Cell Culture

Murine microglia cells (BV-2) were maintained in DMEM-10 (Dulbecco’s Modified Eagle Medium (Thermo Scientific, Waltham, MA, USA), supplemented with 10% fetal calve serum (C-C-Pro)) with standard additives (2mM L-glutamine (Biozym, Hessisch Oldendorf, Germany), 100 µM non-essential amino acids (Biozym), and 100 units/mL penicillin and streptomycin (Biozym)). Pro5 and Lec2 cells were passaged in αMEM5 medium (Modified Eagle Medium α (Thermo Scientific), supplemented with 5% fetal calve serum (C-C-Pro) and standard additives as above.

### 2.5. MNV Cultivation and Titration

MNV-1 (Isolate CW3, passage 6) was cultivated in BV-2 cells in complete DMEM-10 medium and tissue culture infectious dose (TCID)_50_ was determined by end-point dilution in BV-2 cells as described [45].

### 2.6. Production of Lentiviral Vectors and Transduction of CHO Derived Cell Lines

Recombinant lentiviruses for mCD300LF transduction were produced based on the original protocol [46]. Briefly, the gene of interest expressing construct pWPI msc^mCD300LF^ was generated, cloning the murine CD300lf cDNA (Genbank NM_001169153.1, GeneScript) with a C-terminal FLAG-tag between SbfI and MluI restriction sites of pWPI msc GUN (generously provided by Thomas Pietschmann, Twincore, Germany). The recombinant lentivirus was generated transfecting 3 × 10^6^ HEK293T cells with 6 µg pWPI msc^mCD300LF^, 3 µg of the gag/pol packaging vector pCMVΔR8.91 [46], and 1.5 µg of the envelope vector pMD.GVSV-G expressing the vesicular stomatitis virus glycoprotein [47], using polyethylenimine (PEI) as a transfection reagent according to the manufacturer’s protocol (Polysciences, Inc., Warrington, Philadelphia, PA, USA). At 72 h post-transfection, viral supernatants were harvested and filtered (0,45 µm pore-size) to remove aggregates. Transduction of Pro5 and Lec2 cell lines, generously provided by Niklas Arnberg, Umeå University, Sweden, [48] was performed in a 6-well plate incubating confluent Pro5 or Lec2 cells with 1 mL of lentivirus supernatant for 6 to 8 h in the presence of 5 µg/mL polybrene followed by 2 weeks of geneticin (G418) selection. Stable expression of the receptor in Pro5^mCD300LF^ and Lec2^mCD300LF^ cell lines was verified susceptibility to MNV-1 infection. MNV-1 titers yielded up to 2 × 10^8^ TCID_50_ per ml, which is comparable to naturally susceptible cell lines.

## 3. Results

A number of glycans were selected as ligands to probe binding to human and murine norovirus capsid proteins. The glycans in Scheme 1 represent key structural motifs of HBGAs, and sialoglycans. We also included the Galili epitope [49,50] **11**, not present in humans, as well as the Forssman antigen [51] **13** as negative controls that should bind neither to human nor to murine noroviruses [50]. Protein-based chemical shift perturbation (CSP) NMR binding experiments are known to provide unambiguous and highly accurate binding affinities for ligand molecules such as glycans binding to proteins under near-physiological conditions [43]. However, the need to label the proteins with stable isotopes often discourages the use of this method, when proteins are larger than 20 to 30 kDa. Proteins of the size of the P-dimers studied here, with molecular weights around 70 kDa, are considered a challenge. Fortunately, we succeeded in providing high-yield expression protocols for stable isotope labeled HuNoV P-domains before [35,36,52], and recently we established such a protocol also for MNV P-domains (see accompanying paper). For the present study we employed recombinantly expressed and uniformly ^2^H,^15^N-labeled P-domains of selected VP1 capsid proteins. All analyzed P-domains formed dimers (P-dimers) and all HuNoV P-dimers showed the previously reported HBGA binding specificities. To systematically study binding of HBGAs and sialoglycans, we chose P-dimers of previously studied human GII.4 strains, Saga and MI001. For the GII.4 strain VA387, which has been reported to bind sialoglycans [22], only binding of sialoglycans was cross-examined. To study potential MNV-glycan interactions, we selected the MNV-1 CW1 strain, for which infectiousness has previously been shown to depend on sialic acid [19], and an MNV strain isolated in 2007 (MNV07). CSP NMR experiments were complemented by saturation transfer difference (STD) NMR [44] experiments using unlabeled recombinant P-dimers and human GII.10 Vietnam 026 virus-like particles (VLPs).

### 3.1. Chemical Shift Perturbation (CSP) NMR Experiments Provide Accurate Dissociation Constants for HBGAs Binding to Human GII.4 SAGA and MI001 P-Dimers

We employed TROSY HSQC based CSP NMR experiments to study binding of HBGAs to uniformly ^2^H,^15^N-labeled P-dimers. For GII.4 Saga P-dimers an almost complete backbone assignment is available [35]. As the HuNoV P-dimers studied here share more than 90% sequence identity one may expect assignments to be transferrable from TROSY HSQC spectra of Saga P-dimers. In fact, the overall cross-peak patterns are rather similar. However, assignment transfer is straightforward only in less crowded spectral regions (cf. Appendix A). For the measurement of binding affinities of HBGAs an assignment is not necessary.

From our previous study, we learned that highly specific spontaneous deamidation of the asparagine residue N373 of Saga P-dimers leads to the formation of an iso-aspartate residue in that position (iD373), essentially switching off HBGA binding [35]. Therefore, prior to CSP titrations we checked the deamidation rate of MI001 P-dimers using ion-exchange chromatography, showing that the half-lives at room temperature are essentially identical to the ones of Saga P-dimers [35] (cf. Appendix A). This is important because the duration of CSP titration experiments is in the range of several hours, and the amount of deamidated protein has to be kept at a minimum during that time period. Therefore, to analyze binding of HBGAs to non-deamidated GII.4 Saga and MI001 P-dimers (N/N P-dimers; both monomers carry N in position 373), freshly prepared protein samples were purified by ion-exchange chromatography and immediately subjected to CSP titration experiments (Figure 1). Monitoring CSPs for purified fully deamidated samples of Saga P-dimers (iD/iD P-dimers; both monomers carry iD in position 373) was not time-critical since these samples were stable and did not change over time. Dissociation constants in the low mM range were determined for binding of blood group H-disaccharide, A-trisaccharide, B-trisaccharide, and B-tetrasaccharide to N/N Saga P-dimers, and to N/N MI001 P-dimers. For iD/iD P-dimers dissociation constants could only be estimated since binding was extremely weak, in the high mM range. All data are compiled in Table 1, and representative CSP titration curves are shown in Figure 1c,d.

We prepared the N373D mutant of the GII.4 Saga P-domain, with the advantage that no spontaneous deamidation is possible in this position. At room temperature N373D P-dimers are stable for at least several weeks as shown by ion exchange chromatography (Appendix A), and as qualitatively supported by monitoring ^1^H,^15^N TROSY HSQC spectra of uniformly ^2^H,^15^N-labeled P-dimers over time. The dissociation constant for B-Tri **3** (Figure 1d and Table 1) is slightly smaller than in the case of N/N Saga P-dimers (2.4 vs. 6.7 mM, respectively).

In previous studies we had observed discontinuities in STD NMR titration curves, which we erroneously assigned to cooperative HBGA binding [21,26,27]. In the light of our more recent studies [35] and the data collected here it is clear that there is no cooperativity associated with HBGA binding to P-dimers or VLPs. The reason for the observed discontinuities must be due to other causes. Therefore, we reevaluated our old STD NMR titration datasets [26,27] by simply ignoring the discontinuities in the binding isotherms and applying a simple one-site binding model instead (cf. Appendix A). The dissociation constants obtained from this analysis are shown in Appendix A The dissociation constants for a H-disaccharide and for a B-trisaccharide derivative (entries 7 and 8 in Appendix A) match the results for H-Dis **1** and B-Tri **3** from CSP NMR titrations (Table 1) quite well. In the course of these reevaluations we also performed an additional experiment to elucidate potential crosstalk [53] between binding of glycochenodeoxycholic acid (GCDCA) and binding of B-Tri **3** to GII.10 Vietnam 026 VLPs. We obtained corresponding STD NMR titration curves in the absence and in the presence of saturating amounts of GCDCA [36], showing that GCDCA has no influence on the dissociation constant *K_D_* of B-Tri **3** (Figure 2 and Table 1).

### 3.2. MNV P-Domains Do Not Bind to HBGAs or Other Neutral Glycans

To address the question whether MNV P-dimers bind to glycans we subjected a variety of glycan ligands (Table 2) to protein-based as well as ligand-based NMR binding experiments, i.e., CSP and STD NMR experiments, respectively. For CSP experiments, we prepared samples of [*U*-^2^H,^15^N]-labeled samples of MNV P-domains of the strains CW1 and MNV07 as described in detail in an accompanying paper, addressing stabilization and reorientation of MNV P-dimers in the presence of a specific bile acid, GCDCA.

Briefly, without GCDCA, MNV P-domains exist in an equilibrium of monomeric and dimeric species. At higher concentrations above ca. 100 μM, dimers prevail. Addition of GCDCA shifts that equilibrium completely towards the dimeric form and at the same time causes conformational rearrangements within the P-dimers (details are found in the accompanying paper). For the purpose of the present study, it is sufficient to note that binding of glycans was tested in the absence and in the presence of GCDCA. We applied a variety of conditions (Table 2) to probe for interactions of MNV P-dimers with HBGAs or other neutral glycans, but we could neither detect CSPs nor STD effects, demonstrating that MNV P-dimers do not recognize HBGAs or other neutral glycans listed in Table 2 (corresponding overlays of ^1^H,^15^N TROSY HSQC spectra of P-dimers in the absence and presence of glycans as well as STD NMR spectra are compiled in the Appendix A clearly illustrating the negative results).

In the course of these glycan-binding studies, we noticed that MNV P-domains, saturated with GCDCA, specifically bind Ca^2+^ with affinities in the μM range. We also detected binding of Mg^2+^ and Mn^2+^ ions. Bivalent metal ion binding is illustrated in Figure 3. If a ligand sample contains metal ion impurities CSPs can obstruct or fake ligand binding. Therefore, it is important to either quantitatively remove bivalent metal ions prior to chemical shift titration experiments with glycan ligands or to saturate metal ion binding pockets. Otherwise, wrong positive results may be obtained. As an example, traces of Mn^2+^ resulting from chemoenzymatic synthesis of B-Tri **3** using human blood group B galactosyltransferase cause cross peaks in ^1^H,^15^N HSQC TROSY spectra to disappear. After addition of EDTA the peaks recur, clearly demonstrating that the effect is not due to the presence of B-Tri **3**. (cf. Appendix A).

### 3.3. MNV and GII.4 P-Dimers Do Not Recognize Sialic Acid

Earlier studies with MNV showed that attachment to murine macrophages involves sialic acid bearing glycolipids or glycoproteins. Based on mutagenesis experiments a glycan (sialic acid) binding site has been proposed, located in the P-domain in a region corresponding to the HBGA binding site of human GII.4 noroviruses [19,20]. To test this hypothesis, we subjected [*U*-^2^H,^15^N]-labeled and unlabeled MNV P-dimers of the strains CW1 and MNV07 to CSP and STD NMR experiments, respectively, using sialic acid and sialic acid bearing glycans as ligands (Table 2). As GCDCA plays an important role in stabilizing MNV P-dimers, CSP NMR experiments were performed in the absence and in the presence of GCDCA. We also varied the pH and used varying buffer compositions to make sure that the observations do not depend on these conditions. In fact, binding of sialic acid glycans was not observed under any of the settings chosen. To illustrate these negative findings, Figure 4 shows a representative STD NMR spectrum of a sample consisting of MNV07 P-dimers in the presence of saturating amounts of GCDCA and 2 mM 3′-sialyllactose (3′SL) as found in GM3 gangliosides. It is obvious that the only STD signals result from GCDCA. In Appendix A, we compile more STD NMR spectra and TROSY HSQC spectra in the absence and presence of sialoglycans to further illustrate these negative results.

Recent studies reported binding of sialic acid-bearing glycans to HuNoV P-dimers of the GII.4 strains VA387 [22] and MI001 [21]. Therefore, we used [*U*-^2^H,^15^N]-labeled P-dimers of VA387, MI001, and Saga to probe binding of sialoglycans using CSP NMR experiments (Table 3, Figure 4b and Appendix A). To our surprise we did not observe effects of binding under any of the conditions chosen, strongly suggesting that previous positive results were due to experimental artifacts. A study using native MS confirms this conclusion and addresses the underlying experimental problems in full detail (Charlotte Uetrecht, personal communication, manuscript in preparation). In this context, it is interesting to note that published MS based data were not fully consistent. Prior to the report on binding of sialic-acid bearing glycans [22] an earlier study from the same laboratory had stated that neither 3′-sialyllactose nor 6′-sialyllactose bind to VA387 P-dimers [31].

### 3.4. Sialic Acid on CHO Derived Cells Expressing the MNV-1 Entry Receptor Does Not Contribute to Infection

In order to determine the effect of MNV-1 infection, susceptible cell lines lacking sialic acid on their cell surface were infected with MNV-1 (Figure 5). For this, a parental CHO derived cell line (Pro5) and a recombinant Pro5 mutant (Lec2), deficient in transporting CMP-sialic acid into the Golgi compartment [48], were transduced to stably express the MNV-1 entry receptor mCD300LF. The parental Pro5 ^mCD300LF^ and recombinant Lec2 ^mCD300LF^ cells support MNV infection similarly to naturally susceptible murine monocyte derived cell lines. The absence of sialic acid from the cell surface of Lec2^mCD300LF^ cells as described previously [48] was monitored by FITC-conjugated wheat-germ agglutinin staining. Binding was determined inoculating Pro5^mCD^ and Lec2^mCD^ with MNV-1 with an MOI of 5 for 1 h on ice removing the inoculum and determining the TCID_50_ of virions bound the cells. Infection was determined by inoculating respective cells with MNV-1 with an MOI of 0.05 for 1 h on ice removing the inoculum and determining infection between 0 and 72 h post infection by end-point dilution. No significant differences were observed at any analyzed timepoint, suggesting that cell surface sialic acids do not contribute to MNV-1 infection in this system.

### 3.5. Glycan Binding to HuNoV P-Domains Has No Equivalent in MNV P-Dimers

Binding of HBGAs to HuNoV P-domains is well documented by a number of crystallographic studies, by NMR experiments, by mass spectrometry, and by other biophysical techniques, providing a molecular basis for potential explanations of the importance of HBGAs for infection with HuNoVs. However, a convincing molecular model describing a mechanism of how exactly HBGAs promote infection [7,8,33,55,56] in analogy to, e.g., prototypic SV40-ganglioside interactions [57,58,59] is still missing. In this context, it is very likely that additional factors including possibly even a proteinaceous receptor as recently identified for MNV [15,16] contribute to HuNoV. Evaluation of studies addressing MNV/glycan or HuNoV/glycan interactions in combination with recent observations in our laboratory raises concerns about the validity of current data interpretation. One inconsistency emerges with respect to the order of magnitude of HBGA/HuNoV P-domain interactions, which had been described as ranging from low μM to low mM dissociation constants. However, recent CSP NMR experiments [35] suggest real affinities may be up to an order of magnitude weaker. The absence of any structural data for norovirus P-domains complexed with sialic acid or sialoglycans represents another mystery, given that a number of studies imply binding to P-domains of MNV as well as of HuNoV. To resolve these conflicts, we examined representative P-domain/glycan interactions employing CSP and STD NMR experiments. CSP NMR experiments represent a highly sensitive tool for probing binding of ligands to target proteins under near-physiological conditions [43] and deliver a wealth of information ranging from binding topologies to allosteric networks to dissociation constants. Chemical shifts of backbone H^N^ resonances reflect even subtle changes in the protein’s environment such as small changes in pH or buffer composition, giving rise to noticeable CSPs. Even very low-affinity binding can be sensed as reflected by the observation of quantifiable CSPs upon binding of B-Tri **3** to fully deamidated Saga P-dimers (cf. Figure 1d). Therefore, if addition of a ligand molecule to a target protein does not induce measurable CSPs it must be concluded that there is no interaction between that protein and the ligand.

In the case of GII.4 Saga P-dimers, we accomplished an almost complete backbone assignment, allowing identification of ligand binding topologies and affinities based on specific CSPs [35,36]. For the other GII.4 P-dimers used in this study no exhaustive assignment is available yet, but some backbone H^N^ resonances could be identified by comparing to the almost sequence-identical Saga strain (Appendix A). For MNV P-domains the situation is more complicated since monomers and dimers are present in solution, and only addition of GCDCA triggers the formation of stable dimers (see the accompanying paper), likely different from the dimers in the absence of GCDCA. Since no CSPs were observed in any of the TROSY HSQC spectra assignments were unnecessary at this point.

## 4. Discussion and Conclusions

Our results indicate that some of the previous hypotheses regarding glycan recognition by noroviruses need to be revised, as will be briefly discussed in the following.

First, previously reported affinities of HBGAs binding to GII.4 P-dimers are up to an order of magnitude too large (Table 1). A follow up study in our laboratory systematically addressed the binding of HBGAs and other glycans to GII.4 P-dimers, significantly extending the current dataset [60]. In preceding NMR-based studies, we observed discontinuities in STD NMR titration curves leading to the hypothesis of cooperative binding and yielding apparent dissociation constants in the low μM range [21,26,27]. In light of our new systematic NMR binding studies, this hypothesis can no longer be maintained. Reevaluation of our old STD NMR titration data, ignoring the steps and assuming a simple one-site binding model, leads to very similar dissociation constants compared to the CSP-derived data presented here. In another previous STD NMR-based study, sub millimolar dissociation constants had been reported for the binding of L-fucose and citrate to GII.10 P-dimers [28]. The reason simply is that in this case, titrations were stopped at ligand concentrations corresponding to the first “step” in the STD NMR titration curve, that can easily be mistaken as the saturation of binding. Although we have not yet resolved the precise origin of the STD NMR titration curve discontinuities, it appears that bulk effects perhaps involving ligand-ligand interactions are responsible. This is substantiated by the observation that the discontinuities strongly depend on pH and almost disappear under certain conditions, e.g., using BisTris buffer at slightly acidic pH [27]. In the case of MS-derived dissociation constants [29,30,31] there seems to be a fundamental problem due to specific structural features of P-domains. Recent systematic MS studies [61] suggest that the high content of β-sheet regions in P-dimers allows for intercalation of glycans in the gas phase, making elimination of unspecific binding very difficult and leading to false positive detection of binding.

Second, neither HuNoV GII.4 or GII.10 P-domains nor MNV P-domains recognize the analyzed sialylated glycans (cf. Table 2 and Table 3). These observations are supported by previous findings showing that the sialic residue of sialyl Lewis^x^ is not involved in binding to GII.4 VLPs [25] or to GII.9 P-domains [62]. Therefore, it must be concluded that previous MS-based data on binding of ganglioside derived glycans to GII.4 VA387 P-domains [22] are due to the formation of artificial associates of sialoglycans and P-domains in the gas phase after Coulomb explosion [61]. Our NMR experiments unambiguously show that sialic acid or sialic bearing glycans neither bind to VA387 P-dimers nor to other P-domains of GII.4 strains tested here (Table 3). This is in accordance with available biological data providing no hint that sialoglycans promote HuNoV infection. On the other hand, it has been shown that sialyl lacto-*N*-tetraose bearing neoglycoproteins inhibit binding of GII.3 (Chron1) and GII.4 (Dijon) VLPs to HBGA-containing human saliva [63], suggesting that intact capsids may exploit more complex mechanisms for glycan recognition than isolated P-dimers. It remains an open and interesting question how such mechanisms are put into effect.

For MNV, biological evidence implies the involvement of sialoglycans in MNV-1 infection [19,20,64]. At first glance, this cannot be reconciled with the observations presented here. Although it was shown in the past by competition experiments that the MNV-1 P-domain resembles the biologically active form [65], this may not apply to possible sialic acid binding sites. Available crystal structures of the MNV-1 P-domain describe an open (PDB: 3LQ6) and a closed conformation (PDB: 6C6Q, 6E47) with significant structural differences in the A’B’-loop and the C’D’-loop. Ligands such as bile acids and bivalent cations contribute to these structural differences. Here, binding to sialoglycans was neither affected by the presence of bile acid nor by bivalent cations; however, the involvement of as yet unknown co-factors for sialoglycan binding cannot be ruled out. Our cell culture experiments performed here with a novel cell line artificially expressing the entry receptor mCD300LF and devoid of surface sialoglycans further suggest that sialoglycans do not play a major role in MNV-1 binding and infection. This is supported by strain-specific differences in the sensitivity to neuraminidase treatment as shown previously.

Third, MNV P-domains neither bind to HBGAs nor to other glycans studied here (Table 2). Taken together, our data suggest that glycan recognition through P-domains is a distinct feature of HuNoV and absent for MNV.

To conclude, our results essentially redefine the glycan recognition code for norovirus P-dimers and prompt for new experiments on the biological side, eventually decrypting hitherto unknown mechanism of how glycans modulate norovirus infection.

## Data Availability

All data are included in this manuscript and in the Appendix A.

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
