# Peer review of "NMR Experiments Shed New Light on Glycan Recognition by Human and Murine Norovirus Capsid Proteins"

_viruses, 2021, doi:10.3390/v13030416_

Round 1

Reviewer 1 Report

The article presented by Creutznacher and collaborators shows new data in norovirus glycans recognition. Previous reports have shown that human norovirus glycan interactions are key for norovirus infectivity in humans. Early studies reveled that the so-called secretor negative individuals (unable to express the H antigen and H-antigen related glycans [i.e. Lewis b/y and A/B blood group antigens]) in their secretions were naturally protected against norovirus infections. Bio-physical data has been obtained by the research group that presents this article and others demonstrating the physical interaction between specific glycans and certain human norovirus genotypes. The dependency of norovirus glycan interactions has been further confirmed with the use of enteroids where it has been shown that enteroids produced from secretor negative individuals are not susceptible to infection with several norovirus genotypes. Similarly, the interaction between murine norovirus and sialic acid had been shown by the research team members of the present publication.

Major concerns.

Surprisingly the results presented here show that some of the previous results obtained by the research team were artifacts and in the present work they claim that RMN is the proper technology to study those interactions that have not been able to be solved by crystallography. This reviewer has some notions in several bio-physic technologies, including SPR, BLI and crystallography, but is not an expert in RMN.

If the authors are so convinced that their previous results were artifacts this reviewer encourages them to publish addenda or retractions to their previous works to inform to the scientific community about the artifacts.

Minor concerns

This is not the first report showing lack of interaction between human norovirus and glycans. For instance, Carmona-Vicente and collaborators showed that a single mutation in the GII.4 genotype was enough to avoid norovirus-glycans interaction (doi: 10.1128/JVI.01023-16). This mutation is present in all the variants of the GII.4 generated from 2006. This reviewer encourages the authors to include that fact in their discussion. Also relevant information about norovirus glycan interactions has been recently shown by Mary Estes group using enteroids from secretor negative individuals as well as from secretor positive individuals where the FUT2 gen has been mutated showing that still some strains are H antigen dependent (doi: 10.1128/mSphere.01136-20; 10.1074/jbc.RA120.014855; doi: 10.1128/mBio.00251-20).

Author Response

Author’s Reply to Review Report (Reviewer 1)

Surprisingly the results presented here show that some of the previous results obtained by the research team were artifacts and in the present work they claim that RMN is the proper technology to study those interactions that have not been able to be solved by crystallography. This reviewer has some notions in several bio-physic technologies, including SPR, BLI and crystallography, but is not an expert in RMN. If the authors are so convinced that their previous results were artifacts this reviewer encourages them to publish addenda or retractions to their previous works to inform to the scientific community about the artifacts

We agree with the reviewer. This topic requires clarification. A major issue is binding data obtained from native MS. As alluded to in the manuscript, there seems to be a fundamental problem in using native MS to determine dissociation constants for glycans binding to proteins with a high content of b-sheets as this is the case for P-dimers. This problem has not been identified or known before and will be described in detail in a study by the Uetrecht laboratory. To my knowledge, this study is finished by now, and a corresponding manuscript is in preparation in that laboratory. As a follow up, we are planning a review putting norovirus(calicivirus)-glycan binding data from different sources and obtained by different biophysical techniques into perspective. One focus of that review will be the problems encountered when measuring dissociation constants with native MS.

The NMR techniques that we have employed in the present paper are protein based NMR techniques allowing to directly measure perturbations of the protein atoms (backbone and/or side chain methyl groups) under almost physiological conditions. It is very important to note that this technique is a gold standard especially when no binding can be detected. We were very puzzled in the beginning when we had to accept after repeated experiments under various conditions (pH etc) that there is absolutely no binding of sialic acid containing glycans to norovirus P-domains (cf. the attached supplementary material, showing all this “negative” data). This now prompts for new experiments especially into the biology of infection of murine noroviruses, which have been shown to be at least promoted by the presence of sialoglycans. There needs to be another yet unidentified “factor”.

For human noroviruses this matches the non-existences of any biological data hinting to a role of sialoglycans in human norovirus infection very well. However, a prominent publication from the Klassen lab (ref. 20 in our manuscript, Han, L.; Tan, M.; Xia, M.; Kitova, N.; Jiang, X.; Klassen, J. S., Gangliosides are ligands for human noroviruses. J Am Chem Soc 2014, 136 (36), 12631-7) and a follow-up study in our lab (re. 21 in our manuscript, Wegener, H.; Mallagaray, Á.; Schöne, T.; Peters, T.; Lockhauserbäumer, J.; Yan, H.; Uetrecht, C.; Hansman, G. S.; Taube, S., Human norovirus GII.4(MI001) P dimer binds fucosylated and sialylated carbohydrates. Glycobiology 2017, 27 (11), 1027-1037) describe binding of sialoglycans via sialic acid residues to GII.4 P-domains. Whereas the Klassen et al. study solely relied on native MS data our study was based on STD NMR, native MS and Biacore. At the time, it was not known that under these conditions the observed binding of sialoglycans to HuNoV P-dimers is an artifact. If one would repeat the experiments under the same conditions one would obtain the same results again. However, our new protein-based NMR experiments unequivocally demonstrate that there is no binding of sialoglycans to HuNoV P-domains at all.

At the time we had been doing the experiments with sialoglycans as published in Glycobiology 2017 we had no NMR assignment of the GII.4 Saga P-domain, which we only got two years later (Mallagaray, A.; Creutznacher, R.; Dulfer, J.; Mayer, P. H. O.; Grimm, L. L.; Orduna, J. M.; Trabjerg, E.; Stehle, T.; Rand, K. D.; Blaum, B. S.; Uetrecht, C.; Peters, T., A post-translational modification of human Norovirus capsid protein attenuates glycan binding. Nat Commun 2019, 10 (1), 1320). Therefore, we had to rely on so-called ligand-based NMR experiments such as STD NMR, which only allow observation of ligand molecules. The protein stays in the dark. Therefore, the positive answer we got in that 2017 Glycobiology paper must be considered an over-interpretation and would not hold up against repeated experiments. On the other hand, at the time this conclusion was backed by what we thought was clear experimental evidence from MS. Ligand-based NMR binding experiments, in contrast to protein-based NMR binding experiments, can easily lead to false positives for reasons that are due to bulk effects, e.g., formation of molecular associates or aggregates, or protein degradation. With protein NMR we always have a unique protein NMR fingerprint that reacts to even minor changes in the chemical environment and gives us clear indication about the integrity of the protein.

I am explaining all this to show that this is a really complex matter that needs careful and extended retro-analysis, e.g., by a review article. I also think, it is important that the papers describing binding of sialylated glycans to human norovirus P-domains are kept available to the scientific community in their original form because they highlight how scientific knowledge is subject to perpetual change. This is different from cases where experiments have not been performed according to the state of art or in a sloppy fashion, which is clearly not the case. To the contrary, this is an instructive example of how science develops.

This is not the first report showing lack of interaction between human norovirus and glycans. For instance, Carmona-Vicente and collaborators showed that a single mutation in the GII.4  genotype was enough to avoid norovirus-glycans interaction (doi: 10.1128/JVI.01023-16). This mutation is present in all the variants of the GII.4 generated from 2006. This reviewer encourages the authors to include that fact in their discussion. Also relevant information about norovirus glycan interactions has been recently shown by Mary Estes group using enteroids from secretor negative individuals as well as from secretor positive individuals where the FUT2 gen has been mutated showing that still some strains are H antigen dependent (doi: 10.1128/mSphere.01136-20; 10.1074/jbc.RA120.014855; doi: 10.1128/mBio.00251-20).

It is difficult to compare the study by Carmona-Vicente et al. to our data. We have not been aiming at identification of mutations that potentially switch off HBGA recognition. The point was to present quantitative binding data in the form of dissociation constants for representative HBGAs and representative norovirus genotypes. The single point VA378 mutant referred to in the Carmona-Vicente paper is the Q396R mutant (M1), which according to an ELISA assay is concluded to show impaired binding to HBGAs. On the other hand, the double mutant Q396R / N447D (M4) has dramatically increased binding. On the basis of available structural data, I am not able to rationalize this finding. To my opinion this finding needs to be treated with caution as the data referred to originate from ELISA assays with P-particles, which certainly is not "hard" binding data. To clarify this, we are now planning to produce both mutants isotopically labeled and measure affinities using protein based CSP NMR experiments. If there was such pronounced effect on HBGA binding, we would be able to quantify this with the NMR toolbox we now have in hands. In case, the finding can be reproduced in a quantitative fashion this would be very interesting since it would point to yet unidentified allosteric interactions.

For now, we will include a reference to the study mentioned (ref. 10 in our revised manuscript) but we cannot compare it directly to our data as there is a notable gap in the biophysical characterization of binding. The work from the Estes group has already been extensively quoted in our manuscript, in particular the latest work mentioned by reviewer 1 (ref. 11 in our original manuscript, Haga et al., 2020; ref. 12 in the revised manuscript).

Reviewer 2 Report

The authors of this manuscript tried to determine the binding of glycan to Human and murine norovirus capsid using NMR experiments. They determined the dissociation constants for HBGAs binding to Human GII.4 Saga and MI001 P-dimers, however, both CSP and STD NMR experiments failed to prove the interaction between MNV P-dimers and HBGAs and other studied neutral glycans. The results also showed that MNV and GII.4 P-dimers don’t bind with sialic acid.

The study was carefully designed and carried out. The data provides additional knowledge on norovirus recognition mechanism.

Author Response

We would like to thank the reviewer for his positiv review.